# Seed Phytochemical Profiling of Three Olive Cultivars, Antioxidant Capacity, Enzymatic Inhibition, and Effects on Human Neuroblastoma Cells (SH-SY5Y)

**DOI:** 10.3390/molecules27165057

**Published:** 2022-08-09

**Authors:** Irene Gouvinhas, Juliana Garcia, Daniel Granato, Ana Barros

**Affiliations:** 1Centre for the Research and Technology of Agro-Environment and Biological Sciences (CITAB)/Institute for Innovation, Capacity Building and Sustainability of Agri-Food Production (Inov4Agro), University of Trás-os-Montes e Alto Douro, 5001-801 Vila Real, Portugal; 2AquaValor-Centro de Valorização e Transferência de Tecnologia da Água-Associação, Rua Júlio Martins n° 1, 5400-342 Chaves, Portugal; 3Bioactivity and Applications Laboratory, Department of Biological Sciences, Faculty of Science and Engineering, University of Limerick, V94 T9PX Limerick, Ireland

**Keywords:** circular economy, agro-industrial side streams, phenolic compounds, antioxidant agents

## Abstract

This work evaluated the phytochemical composition of olive seed extracts from different cultivars (‘Cobrançosa’, ‘Galega’, and ’Picual’) and their antioxidant capacity. In addition, it also appraised their potential antineurodegenerative properties on the basis of their ability to inhibit enzymes associated with neurodegenerative diseases: acetylcholinesterase (AChE), butyrylcholinesterase (BChE), and tyrosinase (TYR). To achieve this goal, the phenolic composition of the extracts was determined through high-performance liquid chromatography coupled with photodiode-array detection and electrospray ionization/ion trap mass spectrometry (HPLC-DAD-ESI/MS^n^). The antioxidant capacity was assessed by two different methods (ABTS^•+^ and DPPH^•^), and the antineurodegenerative potential by the capacity of these extracts to inhibit the aforementioned related enzymes. The results showed that seed extracts presented a high content of phenolic compounds and a remarkable ability to scavenge ABTS^•+^ and DPPH^•^. Tyrosol, rutin, luteolin-7-glucoside, nüzhenide, oleuropein, and ligstroside were the main phenolic compounds identified in the extracts. ‘Galega’ was the most promising cultivar due to its high concentration of phenolic compounds, high antioxidant capacity, and remarkable inhibition of AChE, BChE, and TYR. It can be concluded that olive seed extracts may provide a sustainable source of bioactive compounds for medical and industrial applications.

## 1. Introduction

Olive oil production is one of the most critical sectors of the agri-food industry in south Europe and the Mediterranean region. It plays a significant role in the economy of these countries, with an essential social impact as well. The growing awareness of the need to reach “zero waste” has driven attention toward the value of co-products. Therefore, research concerning the use of bioactive compounds obtained from olive oil co-products in the development of innovative applications in the cosmetic, pharmaceutical, nutritional, and technological food industries is even more crucial, nowadays. Thus, it is possible to add value to bioactive compounds from co-products that previously were considered waste by improving the efficiency of their use, leading to cost reduction and lower environmental impact [1,2].

The whole olive (*Olea europaea* L.) stone (i.e., the woody endocarp containing the seed) represents a co-product of both the pitted table olive industry and the olive oil industry, based on de-stoning olives before milling in order to improve oil quality, even if this is not such a common practice in the Mediterranean Basin [3]. Several techniques have been studied with the aim of valorizing this co-product, including combustion, fractionation, production of activated carbon, bio-oil, furfural and resins, and extraction of phenolics and other phytochemicals [4,5]. Nevertheless, to the best of our knowledge, the use of olive seeds as a source of antineurodegenerative compounds has not yet been studied.

Antioxidants are chemical compounds that counteract the oxidation of proteins, DNA, and lipids, and have gained attention from both the scientific community and the general public [6]. Oxidative damage is related to the physiopathology of several diseases, such as metabolic syndrome, atherosclerosis, cancer, cardiovascular disease, and brain aging [7]. In this context, it is well known that oxidative stress is a ubiquitous hallmark of neurodegenerative disorders. Moreover, particular variations in the concentration and activity of acetylcholine have also been observed in the Central Nervous System (CNS) [8]. Therefore, a specific treatment based on increasing acetylcholine concentration through the inhibition of acetylcholinesterase (AChE) and butyrylcholinesterase (BChE), which are critical enzymes in the breakdown of acetylcholine, has been considered to be a promising option. Tyrosinase (TYR) converts L-tyrosine to l-DOPA (3,4-Dihydroxy-l-phenylalanine) and oxidizes L-DOPA to form dopachrome, inducing the production of melanin, a pigment that is associated with hyperpigmentation and neurodegenerative disorders, such as Parkinson’s disease [9].

From the health and biocircular perspectives, phenolic compounds recovered from waste streams have been revealed to possess antioxidant, anticholinesterase, and anti-tyrosinase properties. In addition, these compounds have been suggested to protect against neuronal cell dysfunction and cell death [10]. Their importance as natural compounds for the prevention and treatment of several neurological disorders has increased. Hence, olive oil and its co-products could be a promising alternative source for the recovery of phenolic compounds with biological importance and industrial applications [11,12]. To the best of our knowledge, olive seeds have never been investigated for their neurological protection ability, although this bioactivity is well known for olive oil [11,12]. In fact, several polyphenolic compounds have been recognized for their prevention and/or counteracting of neurodegenerative disorders, including flavonoids, stilbenes, and phenolic acids, with this last one being the least studied class. Some phenolics found in olive oil and also in olive seeds have been revealed to possess potential effects against neurodegenerative diseases, including tyrosol and its derivatives, such as oleuropein, mostly provided in the human diet by olive oil. For instance, salidroside, a glucoside derivative of tyrosol, has been revealed to improve memory retrieval, memory acquisition, and enhance mitochondrial biogenesis in rat brain under normoxic as well as hypoxic conditions [13]. Furthermore, tyrosol has demonstrated the potential to protect neurons against AβO neurotoxicity in vitro and ameliorates synaptic disturbance, oxidative stress response, and cognitive impairment in vivo [14]. Luccarini et al. (2014) also demonstrated that oleuropein aglycone counteracts Aβ42 toxicity in the rat brain, indicating anti-aggregation, neuroprotective, and anti-inflammatory activities [15].

Additionally, few reports have been developed concerning the phytoconstituents of olive seeds, and the ones carried out have shown them to be a source of phenolic compounds, essential for the discrimination of cultivars [13]. Additionally, the use of phenolic compounds from olive seed could be a cost-effective alternative to synthetic antineurodegenerative compounds, contributing simultaneously to the olive oil industry’s sustainability and the improvement of co-product management. On the other hand, since life expectancy is increasing, neurodegenerative diseases are also predicted, which will promote the search for natural antineurodegenerative compounds. Therefore, and taking into account the fact that the use of ultrasound-assisted extraction (UAE) is gaining wide acceptance due to several advantages over other conventional and non-conventional methods [16], this work aims to apply the UAE for the first time in olive seeds, from Cobrançosa, ’Galega’, and ’Picual’ cultivars, to extract phenolic compounds, while also studying their antiradical scavenging potential and antineurodegenerative properties. Thus, this study addresses the following hypothesis: olive seeds inhibit acetylcholinesterase, butyrylcholinesterase, and tyrosinase enzymes, showing, for the first time, their potential for use as therapeutic agents in the prevention or amelioration of cognitive decline.

## 2. Results and Discussion

### 2.1. Phenolic Content of Olive Seeds

The phenolic content of olive seeds from ‘Cobrançosa’, ‘Galega’, and ‘Picual’ cultivars was assessed, as presented in Table 1. No significant differences (*p* > 0.05) were found between cultivars with respect to the total phenol content, with values ranging between 11.90 ± 1.56 and 14.71 ± 1.37 mg GA (Gallic Acid) g^−1^. However, regarding *ortho*-diphenols and flavonoid content, ‘Galega’ seed extracts were revealed to have significantly higher concentrations than ‘Cobrançosa’ and ‘Picual’ samples, with 7.05 ± 0.44 mg GA g^−1^ and 2.40 ± 0.18 mg CAT g^−1^, respectively.

The total phenolic content in olive seeds had higher concentrations when compared with those reported in other studies, such as that of Alu’datt et al. (2011), where concentrations between 1.03 and 2.23 mg GA g^−1^ were determined [17]. Additionally, few studies have investigated the total phenolic content of olive seeds, and none have addressed the total *ortho*-diphenol and flavonoid contents. For example, Falcinelli et al. (2018) determined the total phenolic content in olive seeds from the Moraiolo, Leccino, and Arbequina cultivars, obtaining about 4 mg GAE g^−1^. These results were significantly lower than those obtained in the present study [18]. Flores et al. (2018) also determined lower total phenolic contents in Manzanilla cultivar abscisic acid-untreated olive seeds, with values around 0.09 mg GA g^−1^ [19]. This can be explained by the high efficiency of the extraction method employed in our study (UAE), which presents some advantages, including time extraction reduction, lower temperatures, and a lower amount of solvent. However, Silva et al. (2006) determined total phenolic concentrations between 5.9 and 14.9 mg g^−1^ in olive seeds from ten cultivars grown in Portugal, the results of which were within the value range found in the present study [20]. Other authors have also investigated these contents in wastes from the agro-food industry, such as from spent coffee grounds, in which values of total phenols between 4.80 and 54.48 mg GAE (Gallic Acid Equivalents) g^−1^ and between 0.01 and 32.18 mg QE (Quercetin Equivalents) g^−1^ for total flavonoids were obtained, using similar methodologies [21]. These values were in the range of those of our study. Likewise, similar results of total phenols have been reported in several varieties of date palm fruits (flesh and seeds) grown in Morocco (from 1.14 and 68.43 mg GAE g^−1^), as reviewed by Ibourki et al. (2019) [22]. Our research group also determined these parameters in several food and medicinal plants, including leaves, obtaining values between 6.94 and 199.26 mg GA g^−1^, between 26.40 and 391.76 mg GA g^−1^, and from 0.76 to 70.14 mg CAT g^−1^, for total phenol, *ortho*-diphenol, and flavonoid contents, respectively [23]. Similar values have also been found in winery by-products, namely grape stems [24].

The HPLC–DAD–ESI/MS^n^ screening of the three olive seed cultivars studied presented similar chromatograms. Thus, Figure 1 shows a representative chromatogram of the ‘Galega’ cultivar. Six phenolic compounds were identified and recorded at 280 nm: one phenyl alcohol (peak no 1), two flavonoids (peak no 2 and 4), and three secoiridoids (peak no 3, 5, and 6). The standard solutions were also infused in the mass spectrometer separately to obtain MS fragment ions. In the full scan mass spectra, the deprotonated molecular ions [M-H]^−^ of tyrosol, rutin, nüzhenide, luteolin-7-glucoside, oleuropein, and ligostroside were stable and exhibited higher values (Table 2). Phenolic compound identification was based on the search of these ions, the interpretation of its collision-induced dissociation fragments, retention data, and comparison with data found in the literature [20,25,26,27].

Compound 1 showed a deprotonated molecular ion at *m*/*z* 137, exhibiting a loss of radical methoxy (*m/z* 106) and a loss of a water molecule from its molecular ion (*m/z* 119), respectively. This allowed its identification as tyrosol, which is in agreement with the phenolic alcohol profile described in olive seeds [20].

The [M-H]^−^ ion at *m*/*z* 609 was detected, providing fragmentation base peaks at *m*/*z* 301 and 179. The first fragmentation yielded the rhamnosyl-glucoside moiety at *m*/*z* 301 through the neutral loss of 308 mass units. The dissociation of this *m*/*z* ion 301 by splitting the ring C led to the formation of the second fragment, *m*/*z* 179, indicating its correspondence with rutin, which is in agreement with previous reports [28].

Compound 3 has presented a negatively ionized molecule at *m*/*z* 685, showing other fragments to molecular ions at *m*/*z* 1371, *m*/*z* 731 and *m*/*z* 523, allowing its identification as nüzhenide, which is consistent with previous reports [20,29]. The first fragment displayed indicated [2M-H]^-^ of nüzhenide. The fragmentation base peak at *m*/*z* 731 corresponds to its formic acid adduct [M-HCOO]. The fragment ion at *m*/*z* 523 indicates the loss of the glucoside moiety and the respective yield of its aglycon.

Peak no. 4 demonstrated a deprotonated molecular ion at *m*/*z* 447, resulting in a loss of 162 amu, which is consistent with the yield of its aglycone at *m*/*z* 285, corresponding to fragmentation reported for luteolin-7glucoside, as previously reported [20]. Oleuropein, an ester of hydroxytyrosol linked to an elenolic acid with a glucose moiety, was found in the MS data along with its *m*/*z* signal at 539 and fragment ions, including 377, 307, 275, and 223 (compound **5**). This identification was based on literature in which the same precursor ion and fragment ions were also found in olive products [25]. At a retention time of 26.89 min, the deprotonated molecular ion at *m*/*z* 523 presented losses of 162, 494, and 264 amu, providing fragment ions at *m*/*z* 361, 291 and 259, thus allowing its identification as ligstroside [20].

The quantification of phenolic compounds that have previously been identified was assessed by RP-HPLC-DAD by means of their respective standard solutions. According to Table 3, the most abundant compounds were nüzhenide (32.45 mg g^−1^, on average), ligstroside (4.52 mg g^−1^, on average) and rutin (3.08 mg g^−1^, on average). The other three phenolic compounds found—tyrosol, luteolin-7-glucoside, and oleuropein—presented values ranging between 2.84 ± 0.02 and 2.90 ± 0.04 mg g^−1^, between 2.10 ± 0.00 and 2.21 ± 0.00 mg g^−1^, and between 2.35 ± 0.01 and 2.57 ± 0.02 mg g^−1^, respectively.

To the best of our knowledge, Ryan et al. (2003) are the only authors to have quantified olive seeds’ phenolic compounds on the basis of HPLC analysis. These authors described the presence of tyrosol, hydroxytyrosol, oleuropein-aglycone di-aldehyde (3,4-DHPEA-DEDA), oleuropein, verbascoside, and nüzhenide. The concentration of tyrosol (5–40 mg g^−1^), oleuropein (1–64 mg g^−1^) and nüzhenide (1–76 mg g^−1^) were within the concentration ranges determined in the present study. These differences can be explained on the basis that Ryan et al. (2003) used different extraction conditions and solvents, as well as various other extraction techniques and different standards for phenolic quantification [27]. Furthermore, in this research, the cultivar studied was not the same as in our study, since they investigated the cv. Hardy’s Mammoth, an Australian cultivar, that is cultivated in this country. Therefore, environmental conditions are another important factor that can influence and explain these variations.

Other studies have revealed the presence of some of these compounds in olive seeds through identification using mass spectrometry. Nüzhenide and nüzhenide oleoside were identified for the first time in olive seeds by Servili et al. (1999) and Maestro-Durán et al. (1994), respectively [30,31]. Silva et al. (2006, 2010) also identified several phenolic compounds in olive seeds from the ‘Cobrançosa’ cultivar, namely nüzhenide, nüzhenide 11-methyl oleoside, tyrosol, luteolin, lisgostroside, and rutin [20,26]. Alu’datt et al. (2011) identified nine phenolic acids (protocatechuic acid, hydroxybenzoic acid, vanillic acid, caffeic acid, syringic acid, sinapic acid, ferulic acid, *p*-coumaric acid, and cinnamic acid) and three flavonoids (rutin, hesperidin, and quercetin), which were quantified as a percentage from total phenolic contents, based on HPLC peak areas.

### 2.2. Antioxidant Capacity of Olive Seed Extracts

The antiradical scavenging capacity of phenolic olive seed extracts was determined using the ABTS (2,2′-Azino-bis(3-ethylbenzothiazoline-6-sulfonic acid) diammonium salt) and DPPH (2,2-Diphenyl-1-picrylhydrazyl) methods, and the results are presented in Table 1. As expected, ‘Galega’ cultivar samples showed the higher activity compared to the other studied cultivars, with 64.73 ± 3.58 and 15.93 ± 1.50 µmol Trolox g^−1^ being obtained using ABTS and DPPH, respectively. To the best of our knowledge, Falcinelli et al. (2018) are the only authors to have determined the antioxidant capacity of olive seeds using the same methods. These authors obtained activities between 8.00 and 12.00 µmol Trolox g^−1^ and between 0.1 and 12.00 µmol Trolox g^−1^ using the ABTS and DPPH methods, respectively, revealing lower values than those obtained in our study [18]. However, Alu’datt et al. (2011) also investigated variable extraction parameters such as time, temperature, solvent type, sequential extraction of solvents, and the effects of multi-step extraction on the evaluation of olive seeds’ antioxidant potential [17]. Bijla et al. (2021) also determined the antioxidant capacity of spent coffee grounds using the ABTS method (2.11–2.22 mmol TE (Trolox Equivalents) g^−1^) [21], revealing higher values than those presented in our study. In contrast, our findings were in line with other works, namely the one developed by Yu et al. (2021) [23]. Regarding the winery by-products investigated by our research group, both the ABTS and DPPH approaches revealed higher values than those found in olive seeds (0.38–0.73 and 0.24–0.76 mmol Trolox g^−1^, respectively) [24].

### 2.3. Antineurodegenerative Properties of Olive Seed Extracts

The antineurodegenerative activity of olive seed extracts was determined on the basis of their ability to inhibit the activity of AChE, BChE, and TYR enzymes. The inhibitory activities of seed extract against these enzymes are shown in Table 4. ‘Galega’ olive seed extract was the most potent inhibitor of AChE (IC_50_ = 30.68 ± 12.20 µg mL^−1^), BChE (IC_50_ = 51.54 ± 11.81 µg mL^−1^), and TYR (IC_50_ = 27.42 ± 8.23 µg mL^−1^). Extracts of the ‘Cobrançosa’ and ‘Picual’ cultivars were also able to inhibit cholinesterases and tyrosinase, but to a significantly lesser extent than those from the ‘Galega’ cultivar (*p* < 0.05) (Table 4). 

To the best of our knowledge, anti-cholinesterase and anti-tyrosinase have never previously been reported. The strong inhibition capacities revealed by the ‘Galega’ cultivar could be explained by its high content of phenolic compounds. In fact, and as previously indicated, the phenolic compounds present in olive seeds that are also present in olive oil, such as rutin, luteolin-7-glucoside, or ligostroside, seem to be responsible for several antineurodegenetative properties. In this sense, and since the ‘Galega’ samples represented the cultivar with the highest concentration of phenolic compounds found in olive seeds, it was expected that a significant correlation coefficient would be obtained between these parameters. Phenolic compounds have previously been reported to be the main contributor to these activities in other food/co-product matrices, such as in pomegranate extracts, in which moderate and high correlations were found using these enzyme assays [32]. Nevertheless, in [33], no correlation between antineurodegenerative activity and phenolic compounds was found, suggesting a synergistic effect or the presence of other bioactive compounds responsible for this activity [33]. Despite the lack of thorough investigations concerning the use of these specific phenolic compounds for treating the most common neurodegenerative diseases, Alzheimer’s disease (AD) and Parkinson’s disease (PD), this study reveals the potential use of bioactive waste compounds from olive seeds for industrial applications.

Additionally, different parts of olive fruit, including the seed, peel, and especially the pulp, contain oleuropein, demethyloleuropein, and verbascoside, with nüzhenide being detected only in the stone. Some phenolic compounds are present at higher concentrations in other olive oil co-products. However, there are few studies on the biological properties of olive seed extracts, and none on their antineurodegenerative activity, which could also be attributed to the nüzhenide compound. Moreover, nüzhenide has recently been correlated with positive health effects on metabolic diseases, such as diabetes and obesity [34].

### 2.4. Effects of Methanolic Seed Extracts on SH-SY5Y Viability

The selected olive seed cultivars revealed high capacity to inhibit cholinesterase and tyrosinase. Therefore, evaluation of the effect in SH-SY5Y cell viability was also considered to be necessary, namely, with respect to the mitochondrial function (MTT reduction) (Figure 2).

At the concentrations tested (62.50–1000 µg/mL), olive seed extracts did not affect the mitochondrial function. In reality, some studies have demonstrated the protective and comparative benefits of olive phenols and olives as potential treatments or preventatives for AD. On SH-SY5Y cells, it was claimed that olive phenols, namely oleuropein, verbascoside, and rutin, might prevent Alzheimer’s disease by breaking down fibrils and preventing the aggregation of a protein called amyloid beta (Aβ) of senile plaques [35]. Additionally, research supports the neuroprotective properties of rutin and luteolin-7-glucoside, the compounds found in our study, on SH-SY5Y cells, making it a prospective therapeutic option for human clinical trials [36,37].

Regarding the obtained results, it would be worthwhile exploring the neuroprotective properties of olive seed extracts, since these may provide further insights into future research towards the prevention, and treatment of Alzheimer’s and Parkinson’s diseases, based on new natural matrices.

### 2.5. Correlations between the Chemical Composition and Bioactivity of Olive Seed Extracts

Regarding the ABTS and DPPH data, total phenolic content, *ortho*-diphenols, and total flavonoids were significantly (*p* < 0.05) associated with the antioxidant capacity (Table 5). More specifically, tyrosol (r_ABTS_ = 0.750; r_DPPH_ = 0.680), rutin (r_ABTS_ = 0.839; r_DPPH_ = 0.883), luteolin-7-glucoside (r_ABTS_ = 0.839; r_DPPH_ = 0.973), and ligostroside (r_ABTS_ = 0.765; r_DPPH_ = 0.790) were the main contributors to the antioxidant capacity of the olive seed extracts.

With respect to antineurodegenerative activity, total phenolic content, *ortho*-diphenol content, and total flavonoid content were also significantly associated with the inhibition of BChE and TYR (Table 5). Rutin (r_BchE_ = −0.776; r_TYR_ = −0.883), luteolin-7-glucoside (r_BchE_ = −0.939; r_TYR_ = −0.986), and ligostroside (r_BchE_ = −0.730; r_TYR_ = −0.771) were the main phenolic compounds associated with the inhibition of the enzymes. On the other hand, the inhibition of AChE was influenced by *ortho*-diphenols and flavonoids, especially luteolin-7-glucoside (r = −0.674).

The viability of SH-SY5Y cells was enhanced (*p* < 0.05) by the total flavonoid content (r = 0.766), antioxidant capacity measured by the DPPH assay (r = 0.670), and ligostoside (r = 0.708). However, the contents of luteolin-7-glucoside (r = 0.604, *p* = 0.085) and *ortho*-diphenols (r = 0.640, *p* = 0.064) were also marginally associated with higher cell viability. These results showed that the phenolic compounds in olive seed extracts have a non-toxic profile in SH-SY5Y cells at the concentrations tested.

PCA was applied to investigate the multivariate effects of the chemical composition in the bioactivity of olive seed extracts (Figure 3).

In the two-dimensional projection, it is possible to observe a clear differentiation between the cultivars: the ‘Galega’ variety was characterized by high levels of bioactive compounds (e.g., total flavonoids/phenolics, luteolin-7-glucoside, and ligostroside), increased antioxidant capacity, and inhibition of BChE and TYR. On the other hand, although the ‘Picual’ and ‘Cobrançosa’ varieties had higher oleuropein contents, they had lower inhibition of enzymes and antioxidant capacity. All in all, the seeds from the ‘Galega’ variety seem to be a rich source of bioactive compounds for the mitigation of oxidative stress and inhibiting enzymes associated with neurological diseases. Using principal component analysis (PCA) to illustrate the association between the chemical composition and bioactivity of food extracts has been demonstrated to be an accurate and holistic analytical tool (https://www.sciencedirect.com/science/article/abs/pii/S0308814622014868, accessed on 16 June 2022). Therefore, our work supports the combined use of bivariate and multivariate statistical methods to explain how food composition is associated with bioactivities measured by distinct protocols.

Accordingly, in a recent study, in which the interaction between AChE and olive polyphenols, namely oleuropein, ligstroside, verbascoside, oleoside, dimethyl-oleuropein, hydroxytyrosol, and luteolin 7-glucoside, was analyzed, the molecular docking analysis revealed that dimethyl-oleuropein interacts strongly with AChE [38]. Additionally, in 2018, Figueiredo-González et al. studied the AChE and BChE inhibitory potential of phenol-rich extracts from two olive oils, revealing that both samples presented inhibitory activity against these enzymes [39]. In accordance with Edziri et al. (2019), a positive correlation was observed between the IC_50_ values of anticholinesterase activity and the phenolic extracts of some Tunisian olive leaves [40].

Regarding antioxidant activity, most of the research studies demonstrated the influence of phenolic compounds in the antioxidant activity of *Olea europaea* L. extracts. However, few investigated this correlation with individual phenolic compounds, highlighting the need for scientific research into natural compounds found in olive fruit and its by-products.

## 3. Materials and Methods

### 3.1. Sampling

The present work was carried out with olive seeds from three different olive cultivars (‘Galega Vulgar’, ‘Cobrançosa’, and ‘Picual’), from a certified olive grove, at the National Institute for Agricultural and Veterinary Research (INIAV) located in Elvas, Portugal (38º54′53.35” N and 7º19′11.06” O) during the 2019 crop season. Five trees of each cultivar, of comparable age and vigor and with even spaces between them, were selected within the same growing area and orchard row to avoid differences in agricultural practices, geographical locations, and weather conditions. Only healthy olive fruits, without any infection or physical damage, were harvested at commercial maturity. For each cultivar, about 300 g of olives were collected per tree, totalling 1.5 kg. The olives were then homogenized and immediately transported to the laboratory, where they were processed After removing the pulp, intact olive seeds were removed by means of a stone-crushing process. The seed was then freeze-dried (VirTis BenchTop Pro, SP Scientific, Warminster, PA, USA) and ground into a fine powder with a blender, followed by hermetic storage in the dark at room temperature until analysis.

### 3.2. Ultrasound-Assisted Extraction (UAE) of Bioactive Compounds from Olive Seeds

The phenolic compound extraction protocol previously described by Lameirão et al. [16] was used, with some modifications. Extraction of phenolic compounds was carried out using an ultrasonic apparatus (VCX 500 Vibra-Cell™, Newtown, CT, USA), using a 13-mm-diameter tip with an amplitude, time, and temperature controller. A 50% amplitude was employed. Extraction was performed for powdered samples (2.5 g) with 50 mL of methanol: water (70:30, *v*/*v*) in the ultrasonic apparatus for 40 min at 70 °C. Afterwards, methanolic extracts were centrifuged at 15,493× *g*, at 4 °C for 15 min (Sigma Centrifuges 2–16 K, Germany) and filtered. Following the extraction process, the extraction solvent was evaporated. All the samples were stored at 4 °C until analysis.

### 3.3. Phenolic Composition

Total contents of phenols, flavonoids, and *ortho*-diphenols were determined according to spectrophotometric methodologies reported previously [41].

The content of total phenolics in olive seed extracts was evaluated using the Folin–Ciocalteu spectrophotometric method, using gallic acid (GA) as standard. The content was determined by adding gallic acid standard or sample, Folin–Ciocalteu and sodium carbonate. The absorbance readings were taken at 750 nm, and the results were expressed in mg of gallic acid per gram of dry weight (mg GA g^−1^ DW).

The content of *ortho*-diphenols in olive seeds was determined by adding sodium molybdate and gallic acid or sample, and the absorbance was read at 375 nm. For the quantification, gallic acid was used as standard. The content of *ortho*-diphenols was expressed in mg of gallic acid per gram of dry weight (mg GA g^−1^ DW).

The total flavonoid content in olive seeds was determined by the aluminum complex method, using catechin as standard. The results are expressed in mg of catechin per gram of dry weight (mg CAT g^−1^ DW). For all analyses, three replicates (*n* = 3) were performed in 96-well microplates (Nunc, Roskilde, Denmark), and the absorbance measurements were performed with the resort to a microplate reader Infinite M200 (Tecan, Grödig, Austria).

The polyphenolic profile of olive seed samples was assessed by Reverse Phase–High-Performance Liquid Chromatography–Diode Array Detector (RP-HPLC-DAD), according to the method previously described [15]. For this, an Agilent HPLC 1100 series equipped with a photodiode array detector, and a mass detector in series (Agilent Technologies, Waldbronn, Germany) were used. The equipment has a photodiode array detector (model G1315B), an autosampler (model G1313A), a binary pump (model G1312A), and a degasser (model G1322A). The HPLC system was controlled using Xcalibur software (Agilent, version 08.03). The chromatographic analyses were performed with a C_18_ column (250 mm × 4.6 mm, 5 µm particle size; ACE, Aberdeen, Scotland), the reverse phase HPLC method based on a polar mobile phase composed by solvent A: H_2_O/HCOOH (99.9:0.1, *v*/*v*) and solvent B: CH_3_CN/HCOOH (99.9:0.1, *v*/*v*). The following linear gradient scheme was used (t in min; %B): (0; 5%), (15; 15%), (30; 30%), (40; 50%), (45; 95%), (50; 95%) and (55; 5%). With the last time (55 min), B has to return to 5% to stabilize and prepare the column for the next sample. The analysis was performed at 25 °C, with a flow rate of 1.0 mL/min and a sample injection volume of 20 µL. All samples were injected in triplicate. For the quantification of tyrosol, rutin, nüzhenide, luteolin-7-glucoside, oleuropein, and ligstroside, the respective standards were used at 280 nm. Tyrosol (98%), rutin (≥94%), luteolin-7-glucoside (≥98%), oleuropein (≥98%), and ligstroside (≥95%) were of chromatographic grade and were acquired from Merck (Merck, Darmstadt, Germany). Nüzhenide (98%) was obtained from ChemDirect (CH, USA).

Concentrations were expressed in mg g^−1^ of dry weight (mg g^−1^ DW). The HPLC system was coupled to an ion trap mass spectrometer (ultra HCT Bruker, Bremen, Germany) equipped with electrospray ionization (ESI), and operated in a negative ion mode. Data acquisition and processing were accomplished using the B.01.03-SR2 software for ChemStation for an LC-3D system from Agilent Technologies (Waldbronn, Germany), as previously described [18]. The capillary and voltage were maintained at 350 °C and 4 kV, respectively. Mass scan and daughter spectra were measured from *m*/*z* 100 to 1500. Collision-induced fragmentation experiments were executed in an ion trap, using helium as collision gas and setting the collision energy at 50% [28].

### 3.4. Antioxidant Capacity Assays

According to [42], the free radical scavenging capacity was determined using the ABTS and DPPH spectrophotometric methods. The ABTS assay was performed by the reaction of ABTS^•+^ and sample or standard; the absorbance readings were taken at 734 nm. For the DPPH method, DPPH^•+^ solution and Trolox standard or sample were added, and the absorbance readings were taken at 520 nm. These assays were also performed using 96-well microplates (Nunc, Roskilde, Denmark) and an Infinite M200 microplate reader (Tecan, Grödig, Austria). The results are expressed in mmol Trolox per gram of dried sample (mmol Trolox g^−1^ DW). All analyses were performed in triplicate (*n* = 3) for each sample [43].

### 3.5. Antineurodegenerative Properties of Olive Seeds

#### 3.5.1. Cholinesterase (AChE and BChE) Inhibition

The inhibition activity of AChE and BChE was measured according to Ellman et al. (1961) [44]. The following components were mixed in the wells of a 96-well microplate: 3 mM DTNB (5,5-dithiol-bis-(2-nitrobenzoic acid), 15 mM substrate acethylthiocholine iodide (ATCI) or butyrylthiocholine *iodide* (BTCI), 100 mM phosphate buffer (pH 8.0), and olive seed extracts at different concentrations. Galantamine (standard inhibitor) was used as a positive control, and buffer without extract as a negative control. AChE (electric eel (EC 3.1.1.7, type V-S) or BChE (from horse serum (EC 3.1.1.8)) (0.28 U mL^−1^) was added, and the absorbance was measured at 405 nm for 5 min. The inhibitory activity was expressed as IC_50_ values (the concentration required to inhibit AChE or BChE activity by 50%).

#### 3.5.2. Tyrosinase (TYR) Inhibition

TYR inhibition was determined using the modified dopachrome method [45]. The following components were mixed with the extracts in the wells of a 96-well microplate: 80 µL phosphate buffer (pH 6.8), 40 µL TYR (mushroom TYR (EC 1.14.18.1), and 40 µL L-DOPA. The absorbance was measured at 475 nm, and the inhibitory activity was expressed as IC_50_ values. Kojic acid was used as a standard.

#### 3.5.3. Effects of Olive Seeds on the Viability of Human Neuroblastoma Cells

Human neuroblastoma cells (SH-SY5Y) from the American Type Culture Collection (LGC Standards S.L.U., Spain) were maintained and grown as a monolayer in culture plastic flasks (75 cm^2^). The culture medium was Dulbecco’s Modified Eagle *Medium* (DMEM), containing 10% *Fetal bovine serum* (FBS), 2% penicillin/streptomycin and 1% non-essential amino acids (NEAA). Cells were kept in an incubator at 37 °C with a humidified atmosphere of 95% air and 5% CO_2_. Cells were washed with Hanks’ Balanced Salt solution (HBS), trypsinized, and subcultured in 96-well plates at 25,000 cells/cm^2^. All assays were performed after confluence. Dried extracts were dissolved and diluted in a medium containing 0.1% (*v*/*v*) Dimethyl sulfoxide (DMSO). The final concentration of DMSO did not affect cellular viability. To determine the effect of the extracts, cell viability was assessed 24 h after exposure by MTT reduction. The mitochondrial function was assessed according to Sousa et al. (2009) [46]. Briefly, following cell treatment with methanolic olive seed extracts (5.0–1000 µg/mL) for 24 h, the medium was removed, and cells were incubated for 30 min at 37 °C with a culture medium containing 0.5 mg/mL of MTT. The solution was removed, and formazan crystals were solubilized with 200 μL of DMSO. The resultant purple solution was spectrophotometrically measured at 570 nm. Data are presented as the MTT reduction percentage of treated cells compared to control. Four independent assays were carried out, each one in triplicate.

### 3.6. Statistical Analysis

The results of the present work are presented as mean (*n* = 3) ± standard deviation (SD). The data were also subjected to variance analysis (ANOVA) and a multiple range test (Tukey’s test) for a *p*-value < 0.05, using IBM SPSS statistics 21.0 software (SPSS Inc., Chicago, IL, USA). Correlation analysis (Pearson’s coefficient, r-value) was performed to understand the effect of the chemical composition on the bioactivity of the olive seed extracts. In addition, principal component analysis (PCA) was used to highlight intrinsic differences and similarities in the chemical composition and bioactivity of the extracts. For correlation and PCA, the methods and procedures followed those described in a previous report [47].

## 4. Conclusions

The present work showed that ultrasound-assisted extracts from olive seeds have a potent antioxidant capacity and also the ability to inhibit AChE, BChE, and TYR. ‘Galega’ presented the best results, showing the influence of cultivar on these activities. The positive relation between phenolic composition and both biological activities suggests that the presence of phenolic compounds may explain the biological effects of the extracts. More importantly, for the tested concentrations, extracts did not show significant cytotoxicity in human neuroblastoma cells (SH-SY5Y) under the experimental conditions. Thus, our results suggest that olive seeds could be used as an essential source of compounds for treating global health problems such as neurodegenerative and metabolic diseases. Nevertheless, detailed knowledge about each phytochemical compound mechanism of action is required.

## Figures and Tables

**Figure 1 molecules-27-05057-f001:**
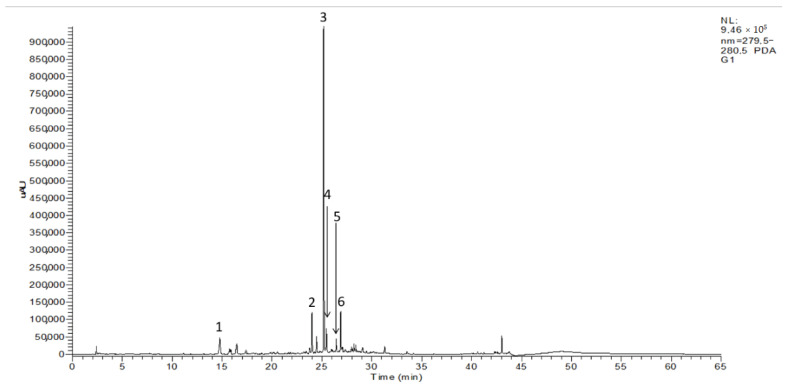
HPLC–DAD chromatogram (280 nm) of olive seeds from ‘Galega’ cultivar: (**1**) tyrosol; (**2**) rutin; (**3**) nüzhenide; (**4**) luteolin-7-glucoside; (**5**) oleuropein; (**6**) ligstroside.

**Figure 2 molecules-27-05057-f002:**
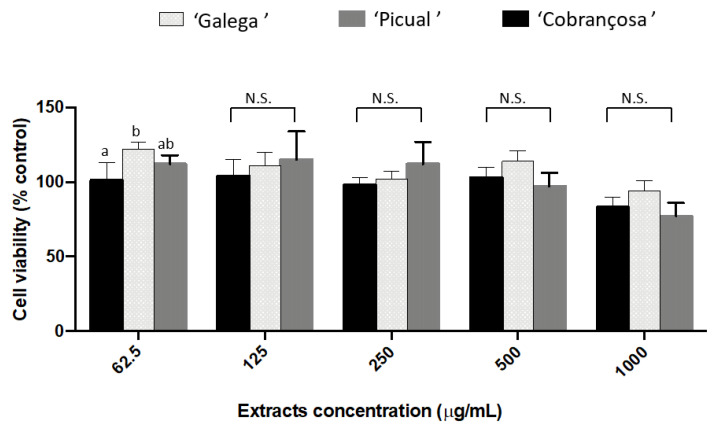
Cytotoxic effect of the olive seeds methanol extracts on SH-SY5Y cells. Data presented as mean (*n* = 3) ± SD values for the same parameter evaluated followed by different superscript lowercase letters are significantly different at *p* < 0.001, according to Tukey’s test. N.S.: not significant.

**Figure 3 molecules-27-05057-f003:**
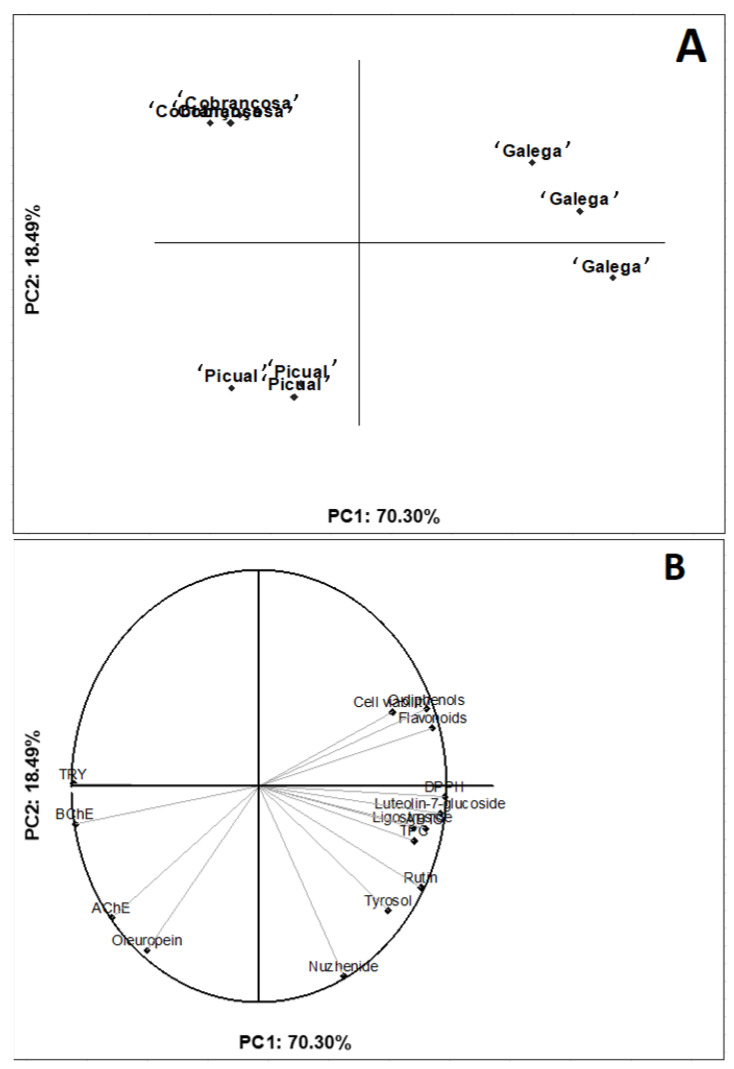
Two-dimensional projection of olive varieties to explain the effects of olive variety on the chemical composition and bioactivity. (**A**) Plot for the olive varieties and (**B**) plot for the responses.

**Table 1 molecules-27-05057-t001:** Total phenol (mg GA (Gallic Acid) g^−1^ DW (Dry Weight)), *ortho*-diphenol (mg GA g^−1^ DW), and flavonoid (mg CAT g^−1^ DW) content and antioxidant capacity (µmol Trolox g^−1^ DW) of olive seeds from different cultivars.

	Phenolic Content			Antioxidant Capacity	
	Total Phenols	*Ortho*-Diphenols	Flavonoids	ABTS	DPPH
‘Cobrançosa’	11.90 ± 1.56 ^a^	4.69 ± 0.16 ^a^	1.50 ± 0.20 ^a^	54.03 ± 3.30 ^a^	3.64 ± 0.18 ^a^
‘GalegaVulgar’	14.71 ± 1.37 ^a^	7.05 ± 0.44 ^b^	2.40 ± 0.18 ^b^	64.73 ± 3.58 ^b^	15.93 ± 1.50 ^b^
‘Picual’	13.03 ± 0.10 ^a^	4.00 ± 0.28 ^a^	1.40 ± 0.05 ^a^	57.42 ± 1.56 ^ab^	6.10 ± 0.55 ^a^
*p*-value	N.S. ^Y^	***	**	*	***

Data presented as mean (*n* = 3) ± SD values for the same parameter evaluated followed by different superscript lowercase letters are significantly different at *p* < 0.001, according to Tukey’s test. ^Y^ Level of significance: N.S.: not significant (*p* > 0.05); * significant at *p* < 0.05; ** significant at *p* < 0.01; *** significant at *p* < 0.001.

**Table 2 molecules-27-05057-t002:** Identification of phenolic compounds in olive seeds from different cultivars by HPLC–DAD–MS^n^ in negative mode.

Peak No.	Compound Id.	RT (min)	λ (UV) (nm)	[M–H]^−^, *m/z*	Fragments
(1)	Tyrosol	14.77	234; 275	137	119, 106
(2)	Rutin	24.00	252; 355	609	301, 179
(3)	Nüzhenide	25.16	248; 275	685	1371, 731, 523
(4)	Luteolin-*7*-glucoside	25.45	256; 350	447	285
(5)	Oleuropein	26.43	245; 280	539	377, 307, 275, 223
(6)	Ligstroside	26.89	249	523	361, 291, 259

**Table 3 molecules-27-05057-t003:** Content of individual phenolics (mg g^−1^ DW) of olive seeds from different cultivars.

Compound	‘Cobrançosa’	‘Galega Vulgar’	‘Picual’	*p*-Value
(1) Tyrosol	2.84 ± 0.02 ^a^	2.90 ± 0.04 ^a^	2.88 ± 0.02 ^a^	N.S. ^Y^
(2) Rutin	2.76 ± 0.04 ^a^	3.35 ± 0.04 ^c^	3.12 ± 0.02 ^b^	***
(3) Nüzhenide	27.75 ± 0.58 ^a^	34.02 ± 1.23 ^b^	35.58 ± 0.21 ^b^	**
(4) Luteolin-*7*-glucoside	2.10 ± 0.00 ^a^	2.21 ± 0.00 ^c^	2.13 ± 0.01 ^b^	***
(5) Oleuropein	2.41 ± 0.04 ^b^	2.35 ± 0.01 ^a^	2.57 ± 0.02 ^c^	***
(6) Ligstroside	3.10 ± 0.05 ^a^	5.45 ± 0.17 ^c^	5.00 ± 0.09 ^b^	***

Data presented as mean (*n* = 3) ± SD values in the same row followed by different superscript lowercase letters are significantly different at *p* < 0.001, according to Tukey’s test. ^Y^ Level of significance: N.S.: not significant (*p* > 0.05); ** significant at *p* < 0.01; *** significant at *p* < 0.001.

**Table 4 molecules-27-05057-t004:** Inhibitory activity of olive seeds extracts against acetylcholinesterase (AChE), butyrylcholinesterase (BChE), and tyrosinase (TYR) (IC_50_, µg mL^−1^).

	AChE	BChE	TYR
‘Cobrançosa’	105.82 ± 17.84 ^b^	158.87 ± 13.55 ^b^	152.61 ± 14.33 ^b^
‘Galega Vulgar’	30.68 ± 12.20 ^a^	51.54 ± 11.81 ^a^	27.42 ± 8.23 ^a^
‘Picual’	186.37 ± 18.61 ^c^	162.11 ± 15.22 ^b^	129.99 ± 16.82 ^b^
Galanthamine	25.03 ± 3.01	21.64 ± 1.22	
Kojic acid			23.45 ± 1.06
*p*-value	*** ^Y^	***	***

Data presented as mean (*n* = 3) ± SD values for the same parameter evaluated followed by different superscript lowercase letters are significantly different at *p* < 0.001, according to Tukey’s test. ^Y^ Level of significance: *** significant at *p* < 0.001. The standard inhibitor was galanthamine for AChE, BChE, and Kojic acid for TYR.

**Table 5 molecules-27-05057-t005:** Correlation analyses between the chemical composition, antioxidant capacity and inhibition of acetylcholinesterase (AChE), butyrylcholinesterase (BChE), and tyrosinase (TYR) of olive seeds extracts.

Responses	Total Phenolic Content	*Ortho*-Diphenols	Flavonoids	ABTS	DPPH	Tyrosol	Rutin	Nuzhenide	Luteolin-7-Glucoside	Oleuropein	Ligostroside	AChE	BChE	TRY	Cell Viability
Total phenols	1.000														
	*p* = ---														
*Ortho*-diphenols	0.669	1.000													
	*p* = 0.049	*p* = ---													
Flavonoids	0.652	0.952	1.000												
	*p* = 0.057	*p* = 0.000	*p* = ---												
ABTS	0.724	0.696	0.706	1.000											
	*p* = 0.027	*p* = 0.037	*p* = 0.034	*p* = ---											
DPPH	0.856	0.863	0.905	0.878	1.000										
	*p* = 0.003	*p* = 0.003	*p* = 0.001	*p* = 0.002	*p* = ---										
Tyrosol	0.767	0.384	0.440	0.750	0.680	1.000									
	*p* = 0.016	*p* = 0.308	*p* = 0.236	*p* = 0.020	*p* = 0.044	*p* = ---									
Rutin	0.796	0.627	0.697	0.839	0.883	0.788	1.000								
	*p* = 0.010	*p* = 0.071	*p* = 0.037	*p* = 0.005	*p* = 0.002	*p* = 0.012	*p* = ---								
Nuzhenide	0.560	0.098	0.205	0.563	0.500	0.750	0.827	1.000							
	*p* = 0.117	*p* = 0.801	*p* = 0.597	*p* = 0.114	*p* = 0.170	*p* = 0.020	*p* = 0.006	*p* = ---							
Luteolin-7-glucoside	0.808	0.853	0.912	0.839	0.973	0.672	0.925	0.576	1.000						
	*p* = 0.008	*p* = 0.003	*p* = 0.001	*p* = 0.005	*p* = 0.000	*p* = 0.047	*p* = 0.000	*p* = 0.104	*p* = ---						
Oleuropein	−0.385	−0.797	−0.695	−0.419	−0.559	−0.006	−0.141	0.418	−0.441	1.000					
	*p* = 0.306	*p* =. 010	*p* = 0.038	*p* = 0.262	*p* = 0.118	*p* = 0.988	*p* = 0.717	*p* = 0.262	*p* = 0.235	*p* = ---					
Ligostroside	0.654	0.564	0.659	0.765	0.790	0.719	0.770	0.500	0.743	−0.360	1.000				
	*p* = 0.056	*p* = 0.114	*p* = 0.053	*p* = 0.016	*p* = 0.011	*p* = 0.029	*p* = 0.015	*p* = 0.171	*p* = 0.022	*p* = 0.342	*p* = ---				
AChE	−0.508	−0.919	−0.878	−0.611	−0.748	−0.194	−0.383	0.185	−0.674	0.943	−0.507	1.000			
	*p* = 0.162	*p* = 0.000	*p* = 0.002	*p* =.080	*p* = 0.020	*p* = 0.618	*p* = 0.308	*p* = 0.634	*p* = 0.047	*p* = 0.000	*p* = 0.164	*p* = ---			
BChE	−0.752	−0.956	−0.963	−0.846	−0.962	−0.541	−0.776	−0.298	−0.940	0.712	−0.730	0.879	1.000		
	*p* = 0.019	*p* = 0.000	*p* = 0.000	*p* = 0.004	*p* = 0.000	*p* = 0.132	*p* = 0.014	*p* = 0.435	*p* = 0.000	*p* = 0.032	*p* = 0.026	*p* = 0.002	*p* = ---		
TRY	−0.796	−0.905	−0.939	−0.871	−0.984	−0.635	−0.883	−0.475	−0.986	0.563	−0.771	0.769	0.980	1.000	
	*p* = 0.010	*p* = 0.001	*p* = 0.000	*p* = 0.002	*p* = 0.000	*p* = 0.066	*p* = 0.002	*p* = 0.196	*p* = 0.000	*p* = 0.115	*p* = 0.015	*p* = 0.015	*p* = 0.000	*p* = ---	
Cell viability	0.366	0.640	0.766	0.524	0.670	0.399	0.409	0.001	0.604	−0.624	0.708	−0.728	−0.705	−0.653	1.000
	*p* = 0.332	*p* = 0.064	*p* = 0.016	*p* = 0.148	*p* = 0.048	*p* = 0.288	*p* = 0.274	*p* = 0.997	*p* = 0.085	*p* = 0.073	*p* = 0.033	*p* = 0.026	*p* = 0.034	*p* = 0.056	*p*= ---

## Data Availability

Not applicable.

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
