# Peer review of "Seed Phytochemical Profiling of Three Olive Cultivars, Antioxidant Capacity, Enzymatic Inhibition, and Effects on Human Neuroblastoma Cells (SH-SY5Y)"

_molecules, 2022, doi:10.3390/molecules27165057_

Round 1
Reviewer 1 Report
This study showed significant differences in phytochemical composition, antioxidant activity and neuroprotective activity of extracts of olive seeds from 3 cultivars (Cobrançosa, Galega, and Picual), with the Galega cultivar having the highest content of total phenolic compounds and the most bioactive of the three. None of the extracts are toxic to human neuroblastoma cells (SH‐SY5Y), making them potentially useful in the prevention of neurodegenerative diseases. The findings are interesting and the manuscript is well-written except that there are places where sentences are not grammatically correct as in lines 82-84, line 289, and line 353. There is also a need to cite the source of acetylcholinesterase (AChE), butyrylcholines-22 terase (BChE), and tyrosinase (TYR) used in their assays.
Author Response
Point 1: This study showed significant differences in phytochemical composition, antioxidant activity and neuroprotective activity of extracts of olive seeds from 3 cultivars (Cobrançosa, Galega, and Picual), with the Galega cultivar having the highest content of total phenolic compounds and the most bioactive of the three. None of the extracts are toxic to human neuroblastoma cells (SH‐SY5Y), making them potentially useful in the prevention of neurodegenerative diseases.
The findings are interesting and the manuscript is well-written except that there are places where sentences are not grammatically correct as in lines 82-84, line 289, and line 353. There is also a need to cite the source of acetylcholinesterase (AChE), butyrylcholines-22 terase (BChE), and tyrosinase (TYR) used in their assays.
Response 1
We appreciate the reviewer’s comments and insightful considerations. We agree with the reviewer´s decision of the need to have a grammatical revision of our manuscript. As suggested by the reviewer, we put major efforts in the English language editing to correct sentences with ambiguous meanings, uniform the tenses and punctuation and rephrasing sentences whenever necessary. The enzymes source was also added into the main text.
We are grateful for the reviewer’s careful reading and comments of our manuscript.
The manuscript was modified accordingly to the reviewer’s suggestions.
If you need any further assistance or information, please, do not hesitate to contact us.
Sincerely,
Ana Barros
Reviewer 2 Report
Dear authors,
This is an important submission towards Molecules. The results are novel and very interesting. However, I see that the manuscript must be revised carefully to be recommended for publication. I hope my comments and suggestions improve the manuscript. Please find my comments below:
1- There are plenty of typos and language mistakes that must be corrected throughout the manuscript,
2- The title must be rephrased for more clarity and fluency: for example Seed phytochemical profiling of three olive cultivars, antioxidant activity, ......,
3- Cultivar's name must be taken between single quotation marks throughout the manuscript including figures and tables,
4- Results of total phenolics, flavonoids, and antioxidant activity must be compared with other food matrices, biomass, and waste to underline the importance of your outcomes. Here are references that may be added and discussed in the Results section: (https://doi.org/10.1155/2021/5138043; https://doi.org/10.1007/s12649-022-01808-8; etc),
5- Authors must indicate what kind of correlation they did (Pearson, Spearman, etc). Likewise, significance level of correlation coefficients must be added. Here is an important reference to be added to clarify your correlations (10.1016/j.cdc.2021.100772). In Table 5, please present only three decimals for correlations coefficients,
6- Fig. 3 must be revised for more clarity,
7- Please clarify the mathematical and biological meaning of your PCA outcomes,
Recommendation: reconsider after major revisions.
Regards.
Author Response
Reviewer´s comment: This is an important submission towards Molecules. The results are novel and very interesting. However, I see that the manuscript must be revised carefully to be recommended for publication. I hope my comments and suggestions improve the manuscript. Please find my comments below:
Point 1 - There are plenty of typos and language mistakes that must be corrected throughout the manuscript.
Response 1 - We appreciate the reviewer’s comments and insightful considerations. We agree with the reviewer´s decision of the need to have a grammatical revision of our manuscript. As suggested by the reviewer, we put major efforts in the English language editing to correct sentences with ambiguous meanings, uniform the tenses and punctuation and rephrasing sentences whenever necessary.
Point 2 - The title must be rephrased for more clarity and fluency: for example Seed phytochemical profiling of three olive cultivars, antioxidant activity, ......,
Response 2 - In accordance with the reviewer´s suggestion, we have rephrased the title of the manuscript.
Point 3: Cultivar's name must be taken between single quotation marks throughout the manuscript including figures and tables,
Response 3 - The suggestions of the referee have been considered.
Point 4 - Results of total phenolics, flavonoids, and antioxidant activity must be compared with other food matrices, biomass, and waste to underline the importance of your outcomes. Here are references that may be added and discussed in the Results section: (https://doi.org/10.1155/2021/5138043; https://doi.org/10.1007/s12649-022-01808-8; etc),
Response 4 - Thanks for the observations. The results were compared with other matrices, including the ones mentioned by the reviewer.
Point 5 - Authors must indicate what kind of correlation they did (Pearson, Spearman, etc). Likewise, significance level of correlation coefficients must be added. Here is an important reference to be added to clarify your correlations (10.1016/j.cdc.2021.100772). In Table 5, please present only three decimals for correlations coefficients,
Response 5 - Thanks for the observations. As already indicated in the Materials and Methods, Pearson’s correlation coefficients were calculated. In the Results section, all p-values are already shown. The decimals were corrected as suggested.
Point 6 - Fig. 3 must be revised for more clarity.
Response 6 - We appreciate the reviewer’s comments and insightful considerations. PCA data are presented in terms of variables and samples. As such, Figure 3 contains both representations in a 2-dimensional plot.
Point 7 - Please clarify the mathematical and biological meaning of your PCA outcomes.
Response 7 - Thanks for the suggestions. PCA data were better exploited and explained in the revised version.
We are grateful for the reviewer’s careful reading and comments of our manuscript.
The manuscript was modified accordingly to the reviewer’s suggestions.
If you need any further assistance or information, please, do not hesitate to contact us.
Sincerely,
Ana Barros
Reviewer 3 Report
The manuscript represents an interesting study about the characterization of bioactive compounds of olive seed, their antioxidant capacity and also their neuroprotective effects.
The work is well formulated and correctly written. However, it is necessary to consider some points and/or improvements.
Line 28: It is not necessary to define the parameters again, they were already defined in lines 22-23
In the introduction, it is necessary to provide more information, about which phenolic compounds have been attributed anti neurodegenerative activities and if this is consistent with the reported profiles of olive seeds.
Please, incorporate a hypothesis at the end of the introduction.
In the results and in Table 1, please express the ABTS and DPPH results as umol/g. Define GA, DW, ABTS, DPPH
Line 97: Please, define GA
Line 125: The molecular ions do not need to be reported here.
Lines 161-162: average values?
Lines 174-175: Could be the variations attributed of the cultivar or the environment? The evaluated cultivars were not the same as in this work
Why are the reported values for total phenol concentrations less than the sum of the individual compounds detected by HPLC? This is not consistent, considering that the Folin method detects phenolic compounds in addition to other compounds. Please explain.
Why are the values detected by the total flavonoid method lower than the rutin concentrations? Please explain.
Line 189: Please, define ABTS and DPPH
Line 212-213: Please, explain this sentence with the correlation coefficient
Figure 2: The statistical treatment is missing in the figure (Tukey)
Line 274: The value 0.667 does not match with what is reported in table 2
In point 4.1, please indicate the amount of fruit sampled on each tree. Was this homogenized?
Lines 331-332: Was this ultrasonic extraction process previously optimized?
Line 335: Why were the extracts stored at 4°C? the degradation of phenolic compounds in methanolic extracts at that temperature is known.
Line 371: Please, indicate the standards used in the quantifications, also their supplier and % purity
Line 380: The reference is missing
Please, homogenize the tables. The five reported tables have different formats
Author Response
Reviewer´s comment: The manuscript represents an interesting study about the characterization of bioactive compounds of olive seed, their antioxidant capacity and also their neuroprotective effects.The work is well formulated and correctly written. However, it is necessary to consider some points and/or improvements.
Point 1 - Line 28: It is not necessary to define the parameters again, they were already defined in lines 22-23
Response 1: We are grateful for the referee´s insightful consideration. We have deleted the repeated information.
Point 2: In the introduction, it is necessary to provide more information, about which phenolic compounds have been attributed anti neurodegenerative activities and if this is consistent with the reported profiles of olive seeds.
Response 2: We agree with the reviewer´s suggestion and the introduction has been improved according to the reviewer comments.
Point 3 - Please, incorporate a hypothesis at the end of the introduction.
Response 3: In accordance with the reviewer´s suggestion, a hypothesis has been added at the end of the introduction.
Point 4: In the results and in Table 1, please express the ABTS and DPPH results as umol/g. Define GA, DW, ABTS, DPPH
Response 4: According to reviewer´s suggestion, we have changed the units and defined these abbreviations.
Point 5: Line 97: Please, define GA
Response 5: Defined.
Point 6: Line 125: The molecular ions do not need to be reported here.
Response 6: As recommended by the reviewer, the molecular ions have been deleted.
Point 7: Lines 161-162: average values?
Response 7 - We appreciate the comment, however, we started with general results and we averaged the three values (from the three cultivars). In this sense, we introduced this information accordingly.
Point 8: Lines 174-175: Could be the variations attributed of the cultivar or the environment? The evaluated cultivars were not the same as in this work
Response 8: In fact, the variations can also be attributed to the cultivar, as well as the environment, since the cultivar is not the same as in our study, and the cultivation country in Australia. These factors have been also added into the manuscript.
Point 9: Why are the reported values for total phenol concentrations less than the sum of the individual compounds detected by HPLC? This is not consistent, considering that the Folin method detects phenolic compounds in addition to other compounds. Please explain. Why are the values detected by the total flavonoid method lower than the rutin concentrations? Please explain.
Response 9: We appreciate the reviewer comment. Spectrophotometric determinations are a rough estimation of the total phenolic content, including all subclasses (e.g. o-diphenols, flavonoids). In fact, the use of different calibrators (gallic acid or catechin for example) plays a huge role in the semi-quantification using the methods, but the results are expressed in Equivalents, it means that these spectophotometric methodologies in fact, just gives us an idea of the behaviour of these classes of compounds taking into account the reaction involved in each one. In the HPLC, we have used standards for the identification and quantification of each compound. Thus, HPLC is the best analytical tool to determine the main phenolic compounds in food matrices.
Point 10: Line 189: Please, define ABTS and DPPH
Response 10: Defined.
Point 11 - Line 212-213: Please, explain this sentence with the correlation coefficient
Response 11 - Thank you for the comment. In fact, and as already added into the introduction as suggested by the editor, the phenolic compounds present in olive seeds and that are also present in huge amounts in olive oil, such as rutin, luteolin-7-glucoside, or ligostroside, seem to be the responsible for several antineurodegenetative properties. In this sense, and since ‘Galega’ samples was the cultivar with the highest concentration of phenolic compounds found in olive seeds, it is expected to obtain a significant correlation efficient between these parameters. This information has been added into the manuscript.
Point 12: Figure 2: The statistical treatment is missing in the figure (Tukey)
Response 12: According to reviewer´s suggestion, we have introduced the statistical treatment of the cell viability test.
Point 13 - Line 274: The value 0.667 does not match with what is reported in table 2
Response 13 - We thank the reviewer remark and, in fact, by mistake, the value was not correct. This has been corrected.
Point 14: In point 4.1, please indicate the amount of fruit sampled on each tree. Was this homogenized?
Response 14: We appreciate the referee´s insightful consideration. The amount of fruit collected from each cultivar was about 1.5Kg (300g per tree) and all the olives were homogenized. This information has been added into the manuscript.
Point 15: Lines 331-332: Was this ultrasonic extraction process previously optimized?
Response 15: We appreciate the reviewer comment. In fact, this ultrasonic extraction process has been widely used and otimized by the research team (Lameirão et al., 2020) with whom our group shares several projects, so this method was also used for this matrix. However, we believe that for future studies it will be convenient to proceed with an adequate optimization, namely through, for example, the Response Surface Methodology that our work team has already been using in the last years.
Point 16: Line 335: Why were the extracts stored at 4°C? the degradation of phenolic compounds in methanolic extracts at that temperature is known.
Response 16: We appreciate the reviewer comment. The extracts were stored at 4 ºC for a short time period (<24h), thus any degradation of phenolic compounds is not significant (<3%, data not shown).
Point 17: Line 371: Please, indicate the standards used in the quantifications, also their supplier and % purity
Response 17: We are grateful for the referee´s insightful consideration. The supplier and purity (%) of each standard have been added into the text.
Point 18: Line 380: The reference is missing
Response 18: The reference has been added.
Point 19: Please, homogenize the tables. The five reported tables have different formats
Response 19: We appreciate the referee´s insightful consideration. The tables have been homogeneized.
We are grateful for the reviewer’s careful reading and comments of our manuscript.
The manuscript was modified accordingly to the reviewer’s suggestions.
If you need any further assistance or information, please, do not hesitate to contact us.
Sincerely,
Ana Barros
Round 2
Reviewer 2 Report
Dear authors,
It seems that the manuscript has been significantly improved and therefore I recommend its publication in Molecules.
Regards.
Reviewer 3 Report
The manuscript has been highly improved with respect to the original submission, including practically all the comments and recommendations.
Just one minor comment:
Line 372: Please, replace Kg by kg, according to the International System of Units.